# Microstructure and Wear Properties of Laser Cladding WC/Ni-Based Composite Layer on Al–Si Alloy

**DOI:** 10.3390/ma14185288

**Published:** 2021-09-14

**Authors:** Xiaoquan Wu, Daoda Zhang, Zhi Hu

**Affiliations:** 1Department of Mechanical Engineerin, Jiangxi Technical College of Manufacturing, Nanchang 330095, China; sdkjdxzdd@126.com; 2Department of Heat Treatment, Jiangxi Institute of Mechanical Science, Nanchang 330002, China; 3City Key Laboratory of Die Surface Treatment & Manufacturing Technology in Nanchang, Nanchang 330095, China; 4School of Mechanical and Electrical Engineering, Nanchang University, Nanchang 330031, China; huzhi@ncu.edu.cn

**Keywords:** Al–Si alloy, laser cladding, WC, wear

## Abstract

The microstructural and wear properties of laser-cladding WC/Ni-based layer on Al–Si alloy were investigated by scanning electron microscope (SEM), X-ray diffraction (XRD), energy dispersive spectrometer (EDS) and wear-testing. The results show that, compared with the original specimen, the microhardness and wear resistance of the cladding layer on an Al–Si alloy were remarkably improved, wherein the microhardness of the layer achieved 1100 HV and the average friction coefficient of the layer was barely 0.14. The mainly contributor to such significant improvement was the generation of a WC/Ni-composite layer of Al–Si alloy during laser cladding. Two types of carbides, identified as M_7_C_3_ and M_23_C_6_, were found in the layer. The wear rate of the layer first increased and then decreased with the increase in load; when the load was 20 N, 60 N and 80 N, the wear rate of layer was1.89 × 10^−3^ mm^3^·m^−1^, 3.73 × 10^−3^ mm^3^·m^−1^ and 2.63 × 10^−3^ mm^3^·m^−1^, respectively, and the average friction coefficient (0.14) was the smallest when the load was 60 N.

## 1. Introduction

Aluminum alloy are widely used in the aviation, aerospace, automobile and other industries due to its low density, high strength and other properties, but the characteristics of poor wear resistance, high temperature and corrosion resistance limits the application of Al alloy [1,2,3].

The surface-modification method is required to make full use of the original properties while improved the surface properties of the material [4,5,6]. Traditional surface modification, including anodizing, electroplating and chemical plating, generates a large amount of wastewater, waste gas and resource consumption, which are not conducive to environmental protection. Additionally, the layer obtained by traditional surface modification is not closely combined with the substrate, and the thickness of the layer is too thin to improve the surface of the alloy [7,8]. Laser cladding is an effective method of surface modification, using a laser to heat both the substrate material and the cladding materials to form a continuous layer on the substrate, that has been widely used in the aviation, aerospace, automobile and other industries. During laser cladding, a protective hard layer can be produced on the surface of the substrate materials to improve the wear resistance and corrosion resistance of various key components in which such alloys are used.

The performance of the layer formed by one kind of cladding material is single, which cannot satisfy the surface performance requirements of most important components. The contituents of ceramics, such as SiC, Al_2_O_3_, B_4_C, WC, have good wear resistance and corrosion resistance, which are usually added into the cladding material to obtain a layer with composite properties. Nickel alloys have high temperature performance and excellent corrosion resistance [9]. Adding Si, Fe, B, C to the nickel powder reduces its melting point. Si, Fe, B, C have strong affinity for and are easy to react with other elements to form a hard phase, which enhances the matrix of a layer using them [10,11]. Adding ceramic to nickel to make composite cladding materials has attracted great attention [12,13,14,15]. A variety of mesophases will be formed in the composite cladding layer because the laser-cladding processis very fast and very hot [16]. These mesophase and ceramic particles affect the properties and wear mechanics of the composite cladding layer.

In view of this, the aim of this work is to prepare the WC/Ni-composite layer on the surface of Al–Si alloy by the laser-cladding-synchronous-powder-feeding method and study the microstructural and wear properties of the layer, which provides a theoretical basis and practical reference for laser cladding on the surface of aluminum alloy.

## 2. Materials and Methods

### 2.1. Experiment Materials

AlSi7Mg alloy with composition: 7.5% Si; 0.5% Fe; 0.2% Cu; 0.25% Mg; 0.25% Ti + Re; Mn < 0.35%; Zn < 0.3% and balance Al (in wt.%) was cut to the size of 100 mm × 50 mm × 12 mm, which was manufactured by Dongguan Panxin metal materials Co., Ltd (Dongguan, China). The roughness of specimens was polished to about 5.5–9.0 μm with sandpaper and cleaned ultrasonically for 15 min.

WC and Ni are spherical powders, which were prepared by China Metallurgical Research Institutes (Beijing, China). the WC particles were about 40–75 μm (as show in Figure 1a) diameter and the Ni-based particles’ diameter were about 45–106 μm (as shown in Figure 1b) and the melting point of Ni-based powder was about 960–1040 °C (composition shown in Table 1). Before being added to the hopper, the WC and Ni powders were mechanically mixed for 30 min.

### 2.2. Experiment Equipment and Methods

The cladding equipment was Nd:YAG solid-state fiber coupling output laser (JHL-GX-2000 laser processing machine, Bangkok, Thailand) with a 1064 nm wavelength, which was produced by Wuhan Chutian Company (Wuhan, China) and a sketch of the equipment is shown in Figure 2a.

The WC/Ni-composite layer was prepared on the surface of the aluminum alloy by synchronous powder-feeding in Ar (25 L/min) configured for 1.2 kw power, 4.6 mm/s scanning speed, 1.5 mm spot diameter, and 20% overlapping.

The microstructure of the samples was characterized by optical microscopy (OM, Nikon, MA-200, Leuven, Belgium) and scanning electron microscopy (SEM, quanta-200, Columbus, OH, USA). The phases in the alloy were analyzed by X-ray diffraction (XRD, D8 ADVANCE, Bruker, Billerica, MA, USA) using Cu Kα radiation in the angular range from 10° to 90°. The microhardness was tested vertically from the surface to the bottom on the cross-section of the layer by a Vivtorinox (HXS-1000A, Shanghai CSOIF Co. Ltd, Shanghai, China) tester for preload weight of 0.2 kg and a preload time of 15 s. Tests collected data three times, at each 100 μm interval, and the average of these data were calculated.

In the pin-on-disc dry-sliding-wear experiment, corundum, with hardness Mohs 9, size Φ40 mm × 10 mm, composition: Fe < 2.5%; Si < 1.2%; Ti < 0.3% and balance Al_2_O_3_ (in wt.%) was selected as material for friction coupling. A sketch of the device’s wear and the processing parameters are demonstrated in Figure 2b and in Table 2. The wear samples were manufactured into size of Φ4.5 mm × 10 mm by wire-electrode cutting followed by polishing with 1000# and 2000# sandpaper. The samples and friction coupling were all soaked in absolute ethanol, and then cleaned ultrasonically.

## 3. Results and Discussion

### 3.1. Microstructure of Multi-Track WC/Ni Layer

The microstructure of the layer was analyzed at a scanning speed of 4.6 mm/s, spot diameter of 1.5 mm and overlap of 20%; the results are depicted in Figure 3. The scanning sequence was from left to right. It is generally accepted that a cladding layer is divided into a cladding zone (CL), transition zone (TZ), and heat-effected zone (HAZ) [17]. Figure 3 shows the layer as divided into CL, TZ, and HAZ, from top to bottom, by dotted lines. A large number of bright, white, spherical WC particles were distributed at the interface between CL and TZ as shown in Figure 3. Cracks were found at the joint of the second and the third cladding (Mark 1), and pores were found at Mark 2 in the HAZ. Due to the metallurgical reaction occurring at the bottom of the layer, or to the protective Ar gas involved in the melting process, the generated or involved gas fails to be discharged in time. Therefore, pores appeared in the heat affected zone at the bottom of the layer, forming porosity defects [18,19].

### 3.2. Composition Analysis of the Layer

There many factors that determined the distribution of composition in the layer, such as convection and mass transfer in the molten pool, the density difference between particles and molten alloy, the interaction between the particles and the solid–liquid interface, the cooling rates of micro regions, the growth direction of dendrites and so on [20]. The microstructure of the top, middle and bottom of the layer is shown in Figure 4a–c. The microstructure of top of layer was composed of white particles and black matrix, as shown in Figure 4a. Due to the large temperature gradient at the top of the layer, it easily formed dense and evenly distribute structures. The zone at the top of the layer was fine-grained [21]. Figure 4b depicts the white spherical particles, the fish-bone and flower-like dendrites that appeared in the middle of layer and at the edge of WC particles, and bright white rod-like grains growing perpendicular to the WC particle boundary. The main reason these appeared was due to the difference of temperate between WC particles and the molten surrounding, causing the WC particles to form a micro-area of directional solidification. Finally, the material surrounding the WC particles formed white, rod-shaped dendrites, which were perpendicular to the edge of WC particles and grew radially [20].

The chemical affinity of tungsten and carbon atoms is not as good as that of Ni, Cr and Fe. At high temperatures, some of tungsten was replaced by Ni, Cr and Fe in a liquid state that combine with carbon atoms to form carbides. In the melt (CL), WC particles reacted with the surrounding Cr, Fe and Ni to form a low-melting-point compound, MxCy [20,21]. The rest of WC was dissolved in the melt to form supersaturated solid solution. With these solidifications, fish-bone and flower-like dendrites were separated out in the melt [22,23]. The bottom of the layer (TZ and HAZ) was composed of coarse columnar crystals, and the grain growth shows obvious directionality, as shown in Figure 4c.

In order to further research the composition of the layer, EDS analysis was carried out for the phases in it, and the atomic percentage of each phase was discerned, as shown in Table 3. The results shown that Al, Ni, Cr and Fe were distributed in each layer of the molten pool. It is indicated that, under the combined action of a large temperature gradient and the Marangoni effect, there was strong convection in the molten pool, leaving the elements of the substrate and cladding materials distributed throughout the molten pool. Subsequently, tungsten was found in the melt, which indicated that WC particles were dissolved in the molten pool. The bright, white particles structure at the top of the layer (CL) were composed of C, Ni, W and Cr. According to the results of EDS, SEM and XRD (Figure 5), M_7_C_3_, M_23_C_3_ were formed in the layer, and Mark B was mainly composed of C and Ni. There was also a solid solution formed by of C, Cr, Al, Si and other elements in melt. The fish-bone dendrites in the middle of the layer (CL) were composed primarily of C, Cr, Ni, W, Al, etc. According to the results of [24] and combined with the results of XRD, the found carbide phase with a cubic crystal structure was (Cr, Ni, Fe)_23_C_6_. The composition of the flower-like dendrites in Mark D was similar to that of Mark C; it was the supersaturated melt, formed by solid solution of C, Cr, W, Fe and other elements in the molten pool, as well as any carbides that precipitated during solidification. The columnar crystal and blocky dendrites, as mark E and mark F shown in Figure 4c were composed of Al, Ni and C. This indicates that melt convection played an important role in the distribution of the elements inside it; indeed, the bulk of the Ni–C moved from the CL to the TZ under the action of melt convection.

Due to the large under-cooling in the molten pool, part of the tungsten, chromium, carbon solid solutes formed supersaturated solid solutions in the melt. According to the Gibbs energy of the solid-state transformation of carbide (Table 4), W–Cr–C–Fe underwent a complex transformation process in the later stages of melt solidification [24,25]:L→M_23_C_6_(S) + γ-(Fe, Ni) (S),(1)
L→M_6_C(S) + WC(S) + γ-(Fe, Ni) (S),(2)
L→M_7_C_3_(S) + WC(S) + γ-(Fe, Ni) (S),(3)

### 3.3. Microhardness Analysis

Figure 6 depicts the microhardness curves of the cross-section of each layer. The microhardness curve of each layer, from surface to bottom, showed obvious stepped descent. The microhardness of the layer was over 1100 HV, and it was about 14 times that of the substrate. The main reason was that some of the WC particles melted into the layer, and forming carbides, strengthening the layer through their dispersion. Secondly, Ni, Cr, W, Si and other elements were dissolved in the melt, which furthered strengthened it. Combined with the above factors, the microhardness of the layer showed a significant improvement.

### 3.4. Wear Test

The wear morphology of the cladding surface, under loads of 20 N, 40 N, 60 N and 80 N, with a friction speed of 0.188 m/s and a friction time of 15 min, was shown in Figure 7. The surface of the layer was characterized by the falling off of the hard phase and the forming of furrows.

When the load was 20 N, there were few and micro-fine scratches on the surface of the layer as shown in Figure 7a, and some small furrows along the tangential direction of WC particles appeared at the edge of WC particles, thus it classified as abrasive wear. When the load increased to 40 N (Figure 7b), deep furrows appeared on both sides of WC particles, which indicated that the wear on the grinding surface was intensified. Cracks appeared on the surface of Mark 2, and its WC particles were penetrated. In Figure 7b, Mark 4 shows a shadow on the edge of its WC particles, along the friction direction. Figure 7c shows that, when the load increased to 60 N, the furrows on both sides of WC particles were deepened, the shadows next to the WC particles were further expanded, and even large pitted areas appeared. Under an 80 N load, as shown in Figure 7d, Mark 1, some of the WC particles were completely broken, and also was classified as abrasive wear. Additionally, bright white blocks were found at Mark 3, shown in Figure 7d. At this stage, adhesive wear may have occurred, and the bright white blocks were corundum friction pairs.

The cracks in the layer were caused by the difference in thermal expansion coefficient between the WC particles and the matrix of the layer; the residual thermal stress of rapid solidification had not been effectively released, which cracked the WC particles and formed cracks in the layer [26]. As shown in Mark 4, a shadow appeared on the edge of the WC particles in the friction’s direction. The WC particles played a supporting role in the layer, easily causing plastic deformations on the contact area of the friction-coupled parts, so the matrix of the layer showed plastic deformation along the wear direction [27]. The shadowy area increased with load. Figure 7e shows the EDS results at Mark 3 and mark 4 in Figure 7b,d. The main elements in the shadow and in the bright white blocks were found to be Al and O. The sources of these elements in the two areas they were found were different. Al and O elements, in the shadowy area, were mainly from friction, and their combined colour was gray. Therefore, when the load was 40 N, there was not only abrasive wear, but also oxidation wear; whereas, Al and O in the bright white blocks were mainly from the components of the friction pair, and the colour was bright white. Therefore, when the load was 80 N, there was not only abrasive wear and oxidative wear, but also adhesive wear occurred.

In order to further analyze the chemical composition of the shadows, line-scanning was carried out in area A of Figure 7b. As shown in Figure 8, the shadow was mainly O. We found that oxides formed on the surface from the friction and increase in temperature. In the process of wear, the oxidation debris entered the shadowed pit. Additionally, the corundum of the friction pair peeled off and was deposited there. Therefore, the content of O and Al in the shadowed pit was increased.

Figure 9 is a schematic diagram of a WC-particle fracture. As wear progressed, most of the pressure and impact of the load was born by the WC particles. These WC particles became more and more prominent, easily forming bumps on the wear surface (Figure 9a). Micro-cracks appeared on the WC-particle surface due to periodic impact. With the extension of friction time, WC particles were completely stripped from the layer, forming pits on the surface of the layer, as shown in Figure 9b (2) and Figure 7c. It is also possible that the WC particles broke with the crack propagation, forming an irregular fracture (as shown in Figure 9b (3) and Figure 7b mark 1).

As per our analysis, the WC particles in the layer had three states: some particles remained in their original spherical shape without any damage, as shown in the particles on the left of Figure 9b. Some particles were completely fractured, forming pits on the surface of the layer (Figure 9b), some were embedded in the layer, and some particles began to crack due to the impact of the countersunk head. The material that was peeled away were left as wear particles between the pairs, as shown in Figure 9b (3).

The formed bumps effectively bore the friction load and reduced the direct contact between the specimen and the friction pair, improving the wear performance of the layer. Additionally, in the cladding process, the hard phase precipitated in the process of powder cooling; this solidification greatly improved the matrix hardness of the layer. The hard phase reduced plastic deformation and the wide spalling of the surface of the layer, also improving the wear performance of the layer.

Figure 10a,b show SEM images of wear debris under the load of 60 N and 80 N, respectively. The found debris was mainly composed of bright white, gray blocks. When the load was 60 N, the debris was fine, with particles about 10 μm in size; at 80 N, the size of debris was 50 μm and bright white. No ductile debris, such as long strips or curls was found, indicating that the debris was mainly brittle fractures of the hard phase. Through EDS analysis of the debris, it was found that the main components of the bright white particles were O and Al, it was indicated that the large, peeled-off debris was mainly corundum, though a small amount of O obtained from the oxides peeled from the oxide film on the grinding surface; the gray block was a mixture of debris from the layer and from the peeling-off of the pair.

Figure 11 shows the wear rate of the specimen under the conditions of room temperature, at a speed of 0.188 m/s, a wear time 15 min and under loads of 20 N, 40 N, 60 N and 80 N.

The wear rate was calculated as follows [27]:(4)$$W_{r}=\frac{\Delta M}{\rho \times L},$$

*W_r_* was the wear rate of the material;

Δ*M* was the mass difference of the sample before and after the test (g);

*ρ* was the density of layer (g/cm^3^);

*L* was the friction stroke (m).

Assuming that the layer did not contain matrix elements, the density of the Ni60 was 8.2 g/cm^3^ and the density of the WC was 15.6 g/cm^3^. According to the mass fraction of the WC added in the mixed powder, the density of the layer was calculated to be 9.67 g/cm^3^. The wear rate of the specimens under different load conditions are shown in Figure 11. According to the change in wear rate, it was found that when the load was 20 N, the wear rate of layer was smallest, only 1.89 × 10^−3^ mm^3^·m^−1^. When the load was 40 N, the wear rate of the layer was 3.42 × 10^−3^ mm^3^·m^−1^, which was 80% higher than under 20 N. At 60 N, the wear rate was 3.73 × 10^−^^3^ mm^3^·m^−1^, an increase of 97% over the 20-N load. When the load was 80 N, the wear rate was 2.63 × 10^−3^ mm^3^·m^−1^, and the wear rate was increased by 39% over that of the 20 N load. We found that, when the load was 20 N, slight scratching occurred on the samples’ surface, and the friction rate was low. When the load reached 40 N, the surface wear of the sample increased, and the wear rate increased significantly. When the load exceeded 60 N, the increase in wear was less. When the load was 80 N, the wear rate did not increase, but rather decreased, compared with that of loads at 40 N and 60 N. The main reason was that, as the load was higher than 40 N, the wear rate of the layer grew steadily with the increase in load. When the load increased to 80 N, the friction pair was severely worn in the wearing process. Thus, some of the cut material was embedded in the surface layer of the substrate (Figure 7d), filling the gaps of the protrusion, such that the wear on the sample was reduced.

The average friction coefficients and transient friction coefficients of the specimens at loads of 20 N, 40 N, 60 N and 80 N are shown in Figure 12. Indicated in Figure 12a, when the load was 20 N and 40 N, the average friction coefficient showed little change, and its values were 0.40 and 0.42. When the load was 60 N, the average friction coefficient decreased to 0.14. When the load increased to 80 N, the average friction coefficient increased sharply, and the value was 0.67. At 60 N, the top of the layer was sharply worn. With the progression of the wear test, the WC bumps that had formed on the surface of the layer bore most of the load, and which was not enough to destroy the WC particles. Therefore, the wear rate was increase. Debris filled the space between the predominant WC particles and the surface of layer; therefore, the coefficient of friction was decreased. Figure 12b–e shows the transient friction coefficients for 15 min under different loads. As shown in the figure, at 20 N (Figure 12b), the amplitude of the transient friction coefficient was the smallest, and when the load was 80 N (Figure 12e), the amplitude of transient friction coefficient was the largest. This indicates that when the load was 20 N, the wear on the specimen was small and the surface remained smooth, so the amplitude of the transient friction coefficient was the smallest. With the increase in load, the wear on the specimen increased. The hardest particles were detached, but remained between the specimen and the opposite grinding pair, making the indirect contact with the surface between the pin and the grinding disc coarser, and, eventually, causing an increase in the amplitude of the transient friction coefficient. When the load increased to 80 N, some WC particles on the pin’s surface were broken, and there were a large number of WC-particle bumps. These factors further increased the roughness of the grinding surface and further increased the amplitude of the transient friction coefficient.

As shown in Figure 12c, when the load was 40 N, the transient friction coefficient and its amplitude were small before 200 s, and maintained their rate of increase; then, after 200 s, the value of transient friction coefficient and its amplitude increased. This was mainly because, at the initial stages of wear, the hardness of the pin’s surface is high, no high-hardness particles fall off, and its surface remains relatively smooth. With the progression of wear, harder particles fall off, leading to an increase in surface roughness, the transient friction coefficient and the amplitude thereof. As shown in Figure 12d, when the load was 60 N, the value of the transient friction coefficient was very stable, the average friction was as low as 0.14, and the fluctuation change was small, showing excellent wear resistance.

## 4. Conclusions

A WC/Ni composite layer on the surface of an Al–Si alloy was obtained by laser cladding. In this layer, a large number of WC particles were distributed at the interface between the CL and TZ. At the edge of the WC particles dissolution occurred and generated the following compounds: AlNi, Al_3_Ni, M_7_C_3_ and M_23_C_3_. The microhardness of the layer had a graded distribution, and the maximum value of microhardness of the layer was about 1100 HV.

Wear-testing indicated that the surface of the layer showed plastic deformation, and wear rate increased with the increase of the load at room temperature for a sliding, dry-friction experiment. When the load higher than 60 N, the corundum materials begun to fall off, and the WC particles in the layer began to break off; when the load reached 80 N, the lost corundum materials were transferred to the layer. At 20 N, 60 N and 80 N, the wear rate of the layer was1.89 × 10^−3^ mm^3^·m^−1^, 3.73 × 10^−3^ mm^3^·m^−1^ and 2.63 × 10^−3^ mm^3^·m^−1^, respectively, and the average friction coefficient (0.14) was the smallest when the load was 60 N.

## Figures and Tables

**Figure 1 materials-14-05288-f001:**
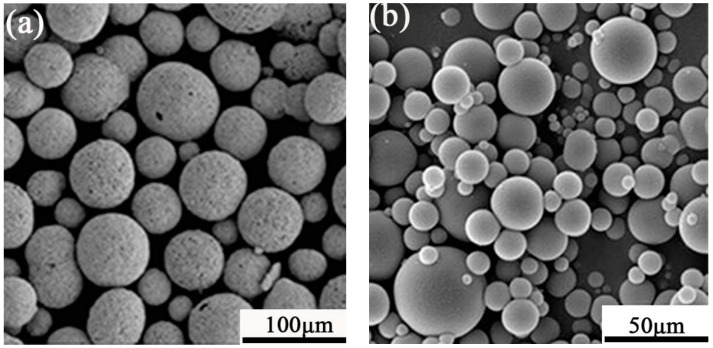
The SEM of (**a**) WC and (**b**) Ni based self-fluxing powder.

**Figure 2 materials-14-05288-f002:**
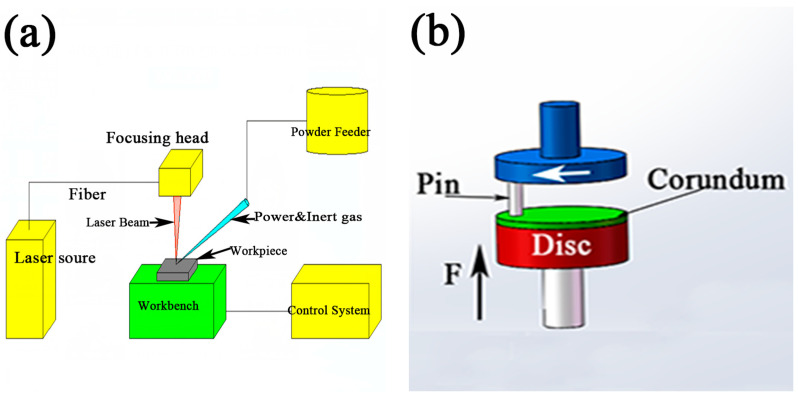
The sketch of (**a**) cladding equipment and (**b**) wear device.

**Figure 3 materials-14-05288-f003:**
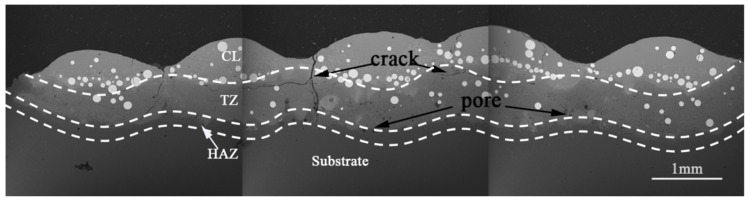
The micromorphology of a multitrack layer obtained with scanning speed of 4.6 mm/s, spot diameter of 1.5 mm and overlap of 20%.

**Figure 4 materials-14-05288-f004:**
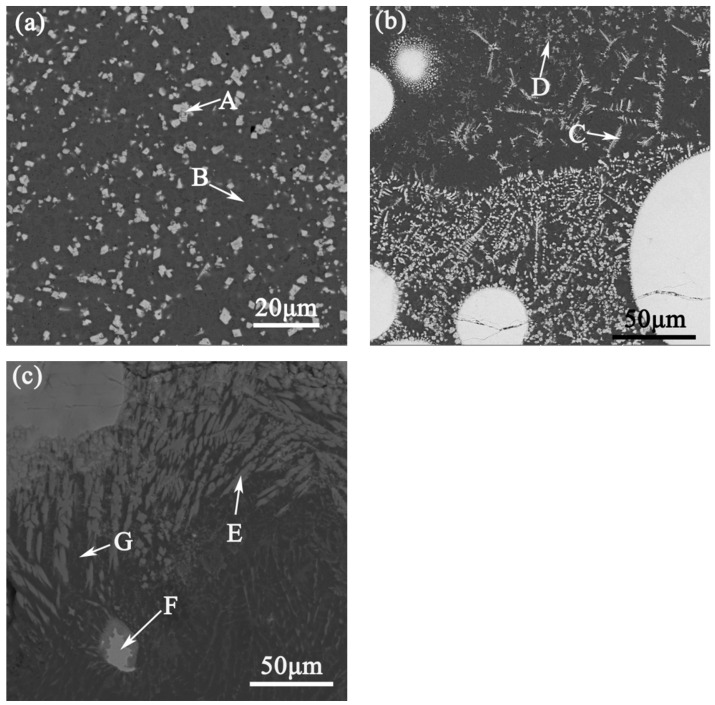
(**a**) Microstructure of the top of CL; (**b**) EBSD in the middle of CL; (**c**) microstructure of TZ and HAZ.

**Figure 5 materials-14-05288-f005:**
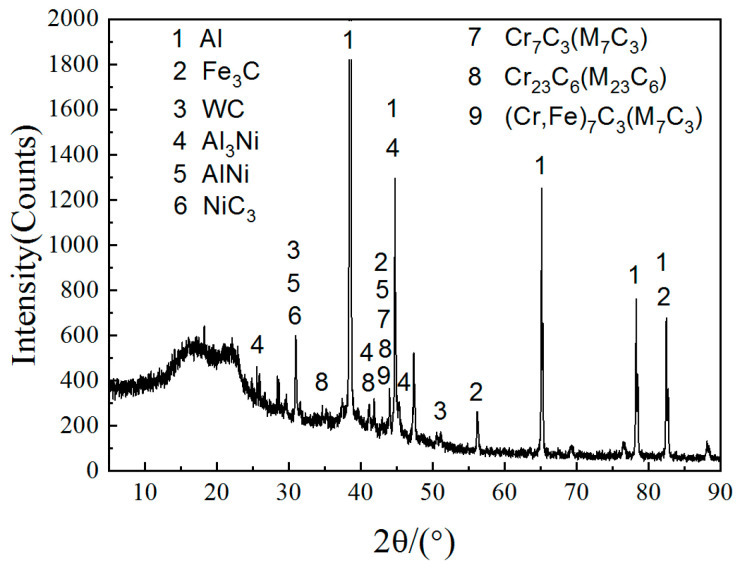
XRD analysis results of the layer.

**Figure 6 materials-14-05288-f006:**
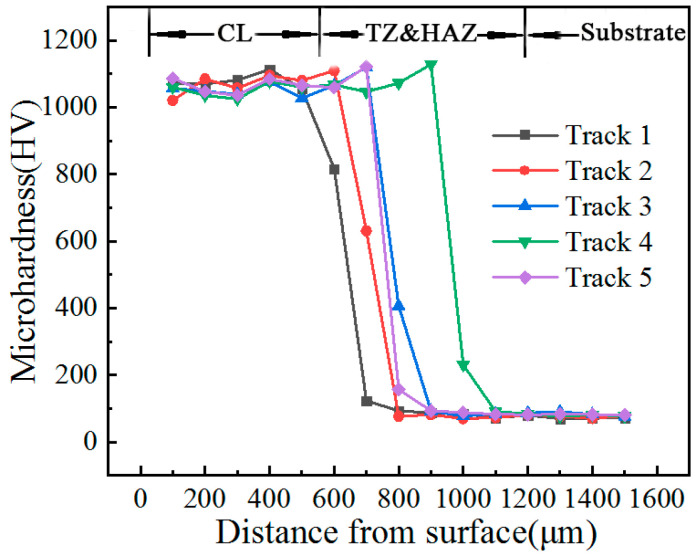
Microhardness curve of cross-section of layer.

**Figure 7 materials-14-05288-f007:**
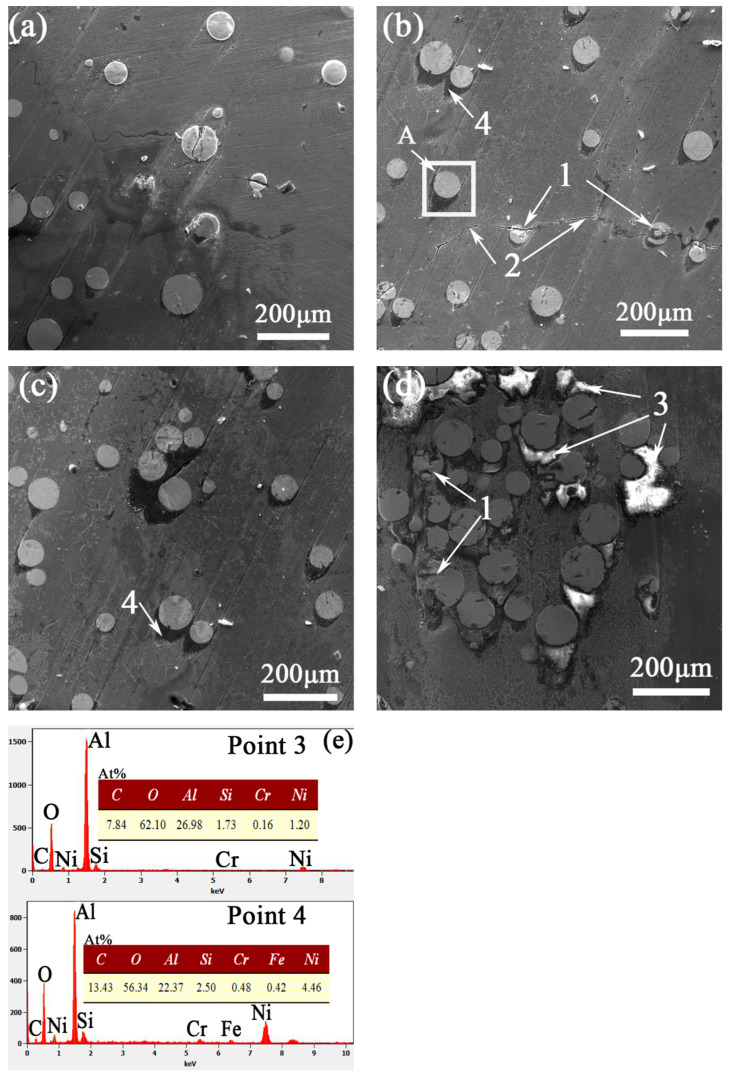
The friction surface, obtained by testing at a speed 0.188 m/s, with a friction time of 15 min and a friction load (**a**) 20 N; (**b**) 40 N; (**c**) 60 N; (**d**) 80 N; (**e**) EDS results at Mark 3 and Mark 4 in the figure.

**Figure 8 materials-14-05288-f008:**
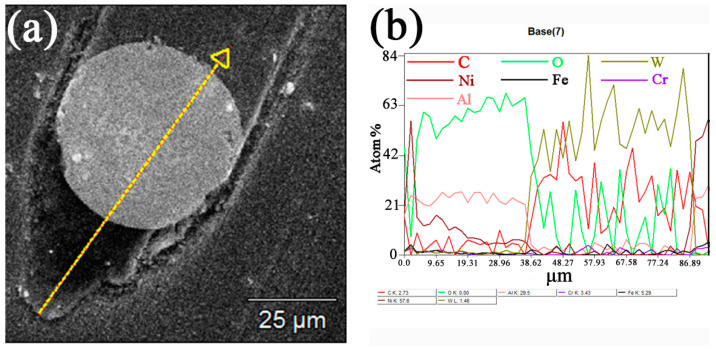
Line-scanning results of area A in Figure 7b. (**a**) SEM of area A; (**b**) the resultant linear element distribution of area A.

**Figure 9 materials-14-05288-f009:**
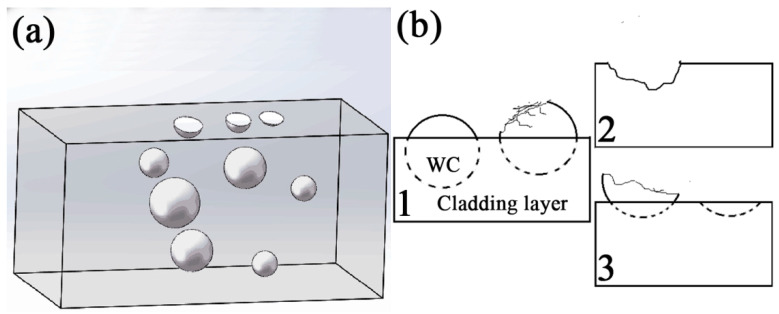
(**a**) A model of a WC-particle fracture; (**b**) the model diagram of 1, 2, 3.

**Figure 10 materials-14-05288-f010:**
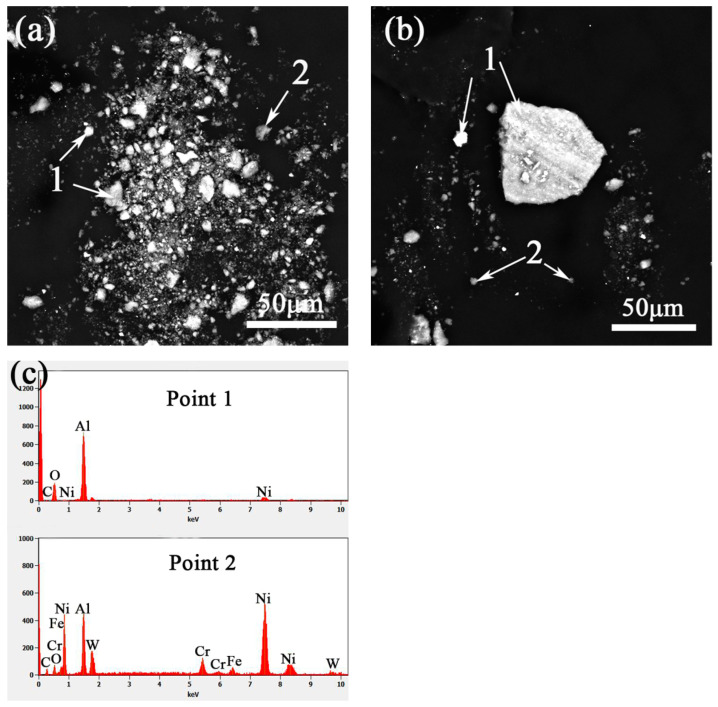
The surface-friction morphology under (**a**) 60 N (**b**) 80 N load; (**c**) results of EDS.

**Figure 11 materials-14-05288-f011:**
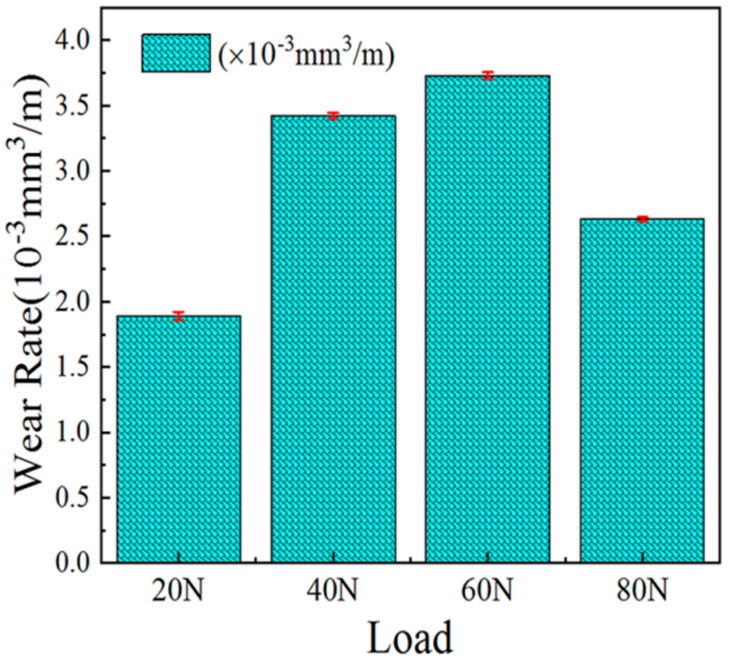
At room temperature, under the condition of 0.188 m/s speed, 15 min wear time and 20 N, 40 N, 60 N, 80 N load, the bar grah of wear rate of sample.

**Figure 12 materials-14-05288-f012:**
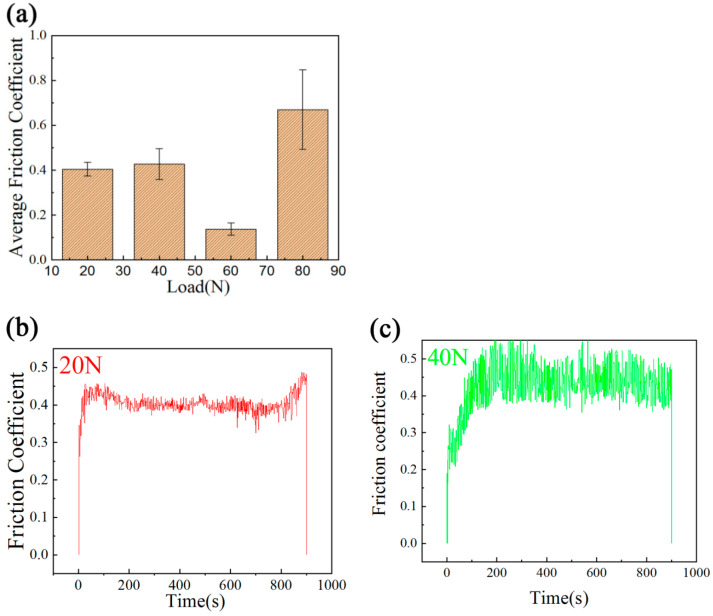
At room temperature, (**a**) the average friction coefficient and the transient friction coefficient of the specimens worn at a speed of 0.188 m/s, for 15 min under loads of (**b**) 20 N; (**c**) 40 N; (**d**) 60 N (**e**) 80 N.

**Table 1 materials-14-05288-t001:** Chemical composition of Ni-based alloy powder (wt.%).

Elements	Cr	Fe	Si	B	C	Ni
Contents	18.0	8.0	4.0	3.2	0.6	bal

**Table 2 materials-14-05288-t002:** The parameters of wear test.

	Rotating Speed (m/s)	Load (N)	Time (s)
Sample 1	0.188	20	900
Sample 2	0.188	40	900
Sample 3	0.188	60	900
Sample 4	0.188	80	900

**Table 3 materials-14-05288-t003:** Atomic percentage of elements in each phase of layer.

Zone		Morphology	Composition (at.%)						
			C	Al	Si	Cr	Fe	Ni	W
**Top of layer**	**A**	Grain	28.81	4.79	–	12.12	3.78	30.92	19.57
	**B**	Matrix	19.08	7.78	7.91	1.96	5.13	58.14	–
**Middle of layer**	**C**	Fish-bone	38.98	10.14	–	11.43	2.52	23.48	18.18
	**D**	Flower-like	36.74	10.14	–	14.75	2.04	15.83	20.50
**Bottom of layer**	**E**	Columnar	15.87	68.49	2.62	0.25	0.46	12.31	–
	**F**	Blocky	35.78	1.10	5.02	4.85	2.11	51.15	–
	**G**	Matrix	17.23	65.38	14.41	–	0.21	2.59	–

**Table 4 materials-14-05288-t004:** The Gibbs energy of carbide transformation [24,25].

Element	Compound	Gibbs Free Energy (∆G·K/mol)
Cr	Cr_23_C_6_	−12,833–3.05 T
Cr	Cr_7_C_3_	−29,985–7.41 T
Cr	Cr_3_C_2_	−9840–2.64 T
Fe	Fe_3_C	−1000 − 7TlnT + 3.5 × 10^−3^ T^2^ − 1.05 × 10^5^ T^−1^ + 46.45 T
Ni	Ni_3_C	8110–1.7 T
W	W_2_C	−7300–0.5 T

## Data Availability

The data is available within the article and can be requested from the corresponding author.

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
