# Peer review of "Microstructure and Wear Properties of Laser Cladding WC/Ni-Based Composite Layer on Al–Si Alloy"

_materials, 2021, doi:10.3390/ma14185288_

Round 1
Reviewer 1 Report
If some of the following contents are reflected or modified, it may be considered for publication in Materials.
- Line No. 48; melting point but also enhance the matrix of layer [10, 11].
-> A detailed explanation is needed for the contents of the 'enhance' the matrix of layer.
- Line No. 63-64;
Was the surface roughness of polished substrates measured before Laser cladding treatment? Depending on the polishing state, the surface roughness may not be the same, even though at the same grinding condition. Depending on the surface roughness, the bonding strength of the clad material and the diffusion distance of atoms from the bonding surface are affected. Therefore, it is necessary to mention the measurement results on the surface roughness.
- Line No. 87; The test were three data at each point with 100μm interval each time.
Although the size of the specimen is small, the number of test measurements is small, so more data results are needed.
- Line No. 89 and 200;
Corundum chemical composition needs to be indicated in the test method. Is the Al from the substrate or from the corundum? Is the corundum pure Al2O3 or does it contain elements such as Cr, Fe, Ti?
- Line No. 106~108;
It is necessary to divide the Substrate, CL, and HAZ areas and mark them in the figure 3. The shape of the matrix and the HAZ contact interface is a kind of waveform. Is it related to #400 polishing or laser condition?
In Fig.3, what is the reason that the cracks develop vertically and horizontally? In particular, transverse cracks will affect subsequent wear tests. Did you take this into account when preparing the wear test specimens?
Cracks occurred in the 1st cladding as well as cracks in the secondary and tertiary cladding. Does this have anything to do with the effect of laser cladding conditions (overlap of 20%)?
- Line No. 120-121;
What part does layer mean? HAZ? CL? or CL+HAZ?
Layer appears frequently in the text, and it is necessary to clearly indicate which part of the layer it is.
- Line No. 165;
It is necessary to distinguish whether the matrix of B and G in Table 3 is the matrix of CL or HAZ.
- Line No. 171-172;
Is the notation of eq.(2) correct?
Is the compound notation for element W in Table 4. correct?
- Line No. 180-181;
Is the layer a CL layer? Why is the hardness of HAZ dropping sharply? In Figure 6, the marked area of HAZ seems to overlap with the substrate, so it is necessary to distinguish it.
- Line No. 224;
In Figure 8 and Figure 10, is the content of O and C reliable by EDS analysis rather than EPMA analysis? The trend of the EDS analysis results presented in Figure 8 is understandable.
- Line No. 253;
The caption of figure 9 needs additional explanation. (b) The model diagram of 1, 2, 3 needs to be improved to make it easier to understand.
- Line No. 189, 276, 279, and 299;
when the -> When the, L or L' ?, the wear -> The wear, and grah -> graph ?
- Line No. 277;
The word ‘layer’ appears frequently in the text, and it is necessary to describe its exact location, e.g., its location in the specimen such as CL, HAZ. If the amount of wear increases due to wear, depending on the specimen, it may start at CL at the beginning but end at HAZ later.
- Line No. 325-327;
The coefficient of friction is affected by the surface roughness of the contact surface. If the surface roughness of the wear test piece was measured before the wear test, the result should be described.
At 60N, the coefficient of friction decreases, but why does the wear rate increase? Although the degree of increase of the abrasion rate is decreased compared to the case of 40N.
- Line No. 334;
It is necessary to supplement the writing by summarizing important parts of the research results in the conclusion.
- Line No. 362-363;
Reference No. 2
- Wu Xiaoquan, Yan Hong, et al. Microstructure and Wear Properties of Ni-based Composite Coatings on Aluminum Alloy Prepared by Laser Cladding. Rare Metal Mat Eng 2020, 49(08): 2574-2582.
I cannot find this reference No. 2 from the website. It is necessary to check whether it is marked correctly.
- Fig. 3, 4, 8;
It can be seen that fine pores are formed at the interface between WC or Ni base powder and CL and HAZ (Fig.3, Fig.4a). Their presence reduces the bonding force between particles and matrix, so it is likely to affect the wear test. And, it is considered that a compound with high hardness such as an intermetallic compound may exist at the interface between particles and matrix during the laser cladding process (Fig. 8b). This can be considered as the cause of the scratch in the tangent direction of the particles during the wear test.
Therefore, it is necessary to show a clear result of analyzing the chemical composition of the phases existing in the diffusion layer or the interface by performing point analysis using EDS by enlarging the contacting or bonding interface.
End.
Author Response
Dear Editor and Reviewers
The revisions suggested by the reviewers are very professional and detailed. We benefit much from these. It is important for us to improve the ability of research. We have revised conscientiously the paper. In manuscript, all significant changes are highlighted in color. The responses to all comments by the editorial office and the referees are provided (point by point) as below:
Reviewer #1:
Line No. 48; melting point but also enhance the matrix of layer [10, 11].
-> A detailed explanation is needed for the contents of the 'enhance' the matrix of layer.
Response: Thanks for your suggestion. Si, Fe,B, C have strong affinity and are easy to react with other elements to form hard phase, which enhances the matrix of layer. It had been revised in manuscript.
Line No. 63-64;
Was the surface roughness of polished substrates measured before Laser cladding treatment? Depending on the polishing state, the surface roughness may not be the same, even though at the same grinding condition. Depending on the surface roughness, the bonding strength of the clad material and the diffusion distance of atoms from the bonding surface are affected. Therefore, it is necessary to mention the measurement results on the surface roughness.
Response: Thanks for your question. The surface of roughness of polished substrates was not measured before laser cladding treatment. The specimens was aluminum alloy, and its laser absorptivity increased with the increase of surface roughness. In this paper, the purpose of polished the surface to increase its surface roughness was to raise the laser absorption of the specimens surface, which is conducive to the formation of molten pool on the substrate surface at the beginning of laser cladding. According to the grade of sandpaper, the roughness Ra of specimens are about 5.5-9.0μm. It had been revised in manuscript.
Line No. 87; The test were three data at each point with 100μm interval each time.
Although the size of the specimen is small, the number of test measurements is small, so more data results are needed.
Response: Thanks for your suggestion. The test were three data at each point with 100μm interval each time and calculated the average of the data for 3 times. The total data of each tracks is 8. It had been revised in manuscript.
Line No. 89 and 200;
Corundum chemical composition needs to be indicated in the test method. Is the Al from the substrate or from the corundum? Is the corundum pure Al2O3 or does it contain elements such as Cr, Fe, Ti?
Response: Thanks for your suggestion. The corundum with composition:Fe<2.5%; Si<1.2%; Ti<0.3% and balance Al2O3(in wt.%) was used in the wear test. It had been revised in manuscript.
Line No. 106~108;
It is necessary to divide the Substrate, CL, and HAZ areas and mark them in the figure 3. The shape of the matrix and the HAZ contact interface is a kind of waveform. Is it related to #400 polishing or laser condition?
Response: Thanks for your suggestion and question. The layer was divided into CL,TZ and HAZ had been shown in Fig.3. Fig.3 had been revised in manuscript. The shape of the layer main depend on the laser condition. In this paper, the laser condition was not change.
In Fig.3, what is the reason that the cracks develop vertically and horizontally? In particular, transverse cracks will affect subsequent wear tests. Did you take this into account when preparing the wear test specimens?
Cracks occurred in the 1st cladding as well as cracks in the secondary and tertiary cladding. Does this have anything to do with the effect of laser cladding conditions (overlap of 20%)?
Response: Thanks for your question. The crack appeared in the layer depend on thermal stress, cladding materials and laser cladding process, etc. In this paper The main reason that the crack appeared in the layer is contact the thermal stress and laser cladding process. In the early stage, our group have been prepared the single-track WC/Ni layer on the Al alloy, and according to the laser condition of single-track determined thelaser condition of multi-track. Our group will study the defect of the cladding layer in the future.
Line No. 120-121;
What part does layer mean? HAZ? CL? or CL+HAZ?
Layer appears frequently in the text, and it is necessary to clearly indicate which part of the layer it is.
Response: Thanks for your question. The layer was divided Cladding Zone(CL), Transition Zone(TZ), and Heat Affected Zone(HAZ) from top to bottom by dotted lines in Fig.3.
Line No. 165;
It is necessary to distinguish whether the matrix of B and G in Table 3 is the matrix of CL or HAZ.
Response: Thanks for your suggestion. Line No.158,167-171,in manuscript have been revised “The main component of columnar crystal and blocky dendrites, as mark E and mark F shown in Fig.4c were Al. and Ni C. It indicated that the melt convection played an important role in the distribution of elements. The bulk Ni-C moved from CL to TZ under the action of melt convection.”
Line No. 171-172;
Is the notation of eq.(2) correct?
Is the compound notation for element W in Table 4. correct?
Response: Thanks for your question. The equation and the table 4 was cited reference to explain the possibility of the reaction. I think it is correct.
Line No. 180-181;
Is the layer a CL layer? Why is the hardness of HAZ dropping sharply? In Figure 6, the marked area of HAZ seems to overlap with the substrate, so it is necessary to distinguish it.
Response: Thanks for your suggestion. The Fig.6 have been revised in the manuscript.
Line No. 224;
In Figure 8 and Figure 10, is the content of O and C reliable by EDS analysis rather than EPMA analysis? The trend of the EDS analysis results presented in Figure 8 is understandable.
Response: Thanks for your suggestion. Because the equipment is limited, EPMA analysis was not added.
Line No. 253;
The caption of figure 9 needs additional explanation. (b) The model diagram of 1, 2, 3 needs to be improved to make it easier to understand.
Response: Thanks for your suggestion. It have been revised in manuscript.
Line No. 189, 276, 279, and 299;
when the -> When the, L or L' ?, the wear -> The wear, and grah -> graph ?
Response: Thanks for your suggestion. The revised have been finished in the manuscript.
Line No. 277;
The word ‘layer’ appears frequently in the text, and it is necessary to describe its exact location, e.g., its location in the specimen such as CL, HAZ. If the amount of wear increases due to wear, depending on the specimen, it may start at CL at the beginning but end at HAZ later.
Response: Thanks for your suggestion. The layer have been describe anew (Line No.111-113) in the manuscript. “ It is generally accepted that the cladding layer was divided into Cladding Zone(CL),Transition Zone(TZ), and Heat Affected Zone(HAZ) [16]. As shown in Fig.3 the layer was divided CL,TZ,and HAZ from top to bottom by dotted lines.”
Line No. 325-327;
The coefficient of friction is affected by the surface roughness of the contact surface. If the surface roughness of the wear test piece was measured before the wear test, the result should be described.
Response: Thanks for your suggestion. In the wear test, the specimens was polished by the sandpaper, the roughness of the specimen is about 0.28μm. This data is an estimate, and its did not appear in the manuscript. Our group will measure the roughness of the specimen before the wear test in the next work.
At 60N, the coefficient of friction decreases, but why does the wear rate increase? Although the degree of increase of the abrasion rate is decreased compared to the case of 40N.
Response: Thanks for your question. When the load is 60N, the top of the layer was sharply worn. With the progress of the wear test, the WC stick out from the surface of the layer to bear the load, and the load was not enough to destroy WC particles. therefore, when the load was 60N, the wear rate was increase. The debris filled the space between the predominant WC particles and the surface of layer. Therefore, the coefficient of friction was decreased.
Line No. 334;
It is necessary to supplement the writing by summarizing important parts of the research results in the conclusion.
Response: Thanks for your suggestion. The conclusion of the paper have been renew in the manuscript.
Line No. 362-363;
Reference No. 2
Wu Xiaoquan, Yan Hong, et al. Microstructure and Wear Properties of Ni-based Composite Coatings on Aluminum Alloy Prepared by Laser Cladding. Rare Metal Mat Eng 2020, 49(08): 2574-2582.
I cannot find this reference No. 2 from the website. It is necessary to check whether it is marked correctly.
Response: Thanks for your question. The reference have been download from the website, and we have been carefully check the information of the paper and the mark in the manuscript. The information of the paper in the manuscript is right. The screenshot of the paper was shown.
Fig. 3, 4, 8;
It can be seen that fine pores are formed at the interface between WC or Ni base powder and CL and HAZ (Fig.3, Fig.4a). Their presence reduces the bonding force between particles and matrix, so it is likely to affect the wear test. And, it is considered that a compound with high hardness such as an intermetallic compound may exist at the interface between particles and matrix during the laser cladding process (Fig. 8b). This can be considered as the cause of the scratch in the tangent direction of the particles during the wear test.
Therefore, it is necessary to show a clear result of analyzing the chemical composition of the phases existing in the diffusion layer or the interface by performing point analysis using EDS by enlarging the contacting or bonding interface.
Response: Thanks for your suggestion. In this work, we can see from Fig.3 and Fig.4a, and found that most of the pore in layer were appeared in the TZ. In the Figure below, the interface between WC and the matrix of the layer have a good bonding, and the surface of WC particle generated some white crystal branch, and the main composition of it is W, Cr, Fe,Ni, etc. According to the reference, the white crystal branch was compound like M6C, M23C6, whose hardness is not as good as WC.
Secondly, the pore in the TZ is the gas, which was generated in the metallurgical reaction or the involved gas fail to discharge in time.

Reviewer 2 Report
The manuscript submitted for review concerns research on the microstructure and wear resistance of Ni coatings reinforced with WC particles. The coatings were made on the Al-Si alloy.
My comments on the manuscript are below:
- The abstract is correct. After reading it, it is clear what the work is about.
- It is possible to compare the hardness of the substrate to the produced coating, but in my opinion it is quite obvious that the hardness has increased.
- The paper can be enriched with other citations concerning composite coatings, e.g. doi.org/10.1016/j.optlastec.2020.106784
- Has the melting point of the nickel alloy been specified correctly? (line 68)
- Enter what was the overlapping of laser paths in the "Experiment equipment and methods" section
- The description of table 2 is incorrect. The table is about the consumption rather than the parameters of the laser cladding.
- Minor typos, sometimes double spaces, such as line 88
- I would rather divide the coating into: Cladding Zone, Melted Zone, Heat Affected Zone and surface. I don't think there would be any WC particles in HAZ. HAZ is just a zone where heat changes the microstructure of the substrate. This zone doesn't melt. I think a different reagent would have to be used to see the microstructure of the substrate. Please consider this in future research.
- Is the determination of NiC3 correct.
- Figure 8a. Do you have better quality for this photo? The photo is very good and you can see the friction mechanism, but better quality would give a beautiful SEM photo effect
- You should expand the "Conclusions" section.
Author Response
Dear Editor and Reviewers
The revisions suggested by the reviewers are very professional and detailed. We benefit much from these. It is important for us to improve the ability of research. We have revised conscientiously the paper. In manuscript, all significant changes are highlighted in color. The responses to all comments by the editorial office and the referees are provided (point by point) as below:
Reviewer #2:
The manuscript submitted for review concerns research on the microstructure and wear resistance of Ni coatings reinforced with WC particles. The coatings were made on the Al-Si alloy.
My comments on the manuscript are below:
The abstract is correct. After reading it, it is clear what the work is about.
It is possible to compare the hardness of the substrate to the produced coating, but in my opinion it is quite obvious that the hardness has increased.
The paper can be enriched with other citations concerning composite coatings, e.g. doi.org/10.1016/j.optlastec.2020.106784
Response: Thanks for your suggestion. The reference have been added in the manuscript.
Has the melting point of the nickel alloy been specified correctly? (line 68)
Response: Thanks for your question. The melting point of the Nickel alloy was provided by the manufacturer and our group have been verified it.
Enter what was the overlapping of laser paths in the "Experiment equipment and methods" section
Response: Thanks for your suggestion. The 20% overlapping have been added in the manuscript.
The description of table 2 is incorrect. The table is about the consumption rather than the parameters of the laser cladding.
Response: Thanks for your suggestion. The description of table 2 have been revised to “The parameters of wear test” in the manuscript.
Minor typos, sometimes double spaces, such as line 88
Response: Thanks for your suggestion. Our group have been carefully check the manuscript, and revised the all the typos in the paper.
I would rather divide the coating into: Cladding Zone, Melted Zone, Heat Affected Zone and surface. I don't think there would be any WC particles in HAZ. HAZ is just a zone where heat changes the microstructure of the substrate. This zone doesn't melt. I think a different reagent would have to be used to see the microstructure of the substrate. Please consider this in future research.
Response: Thanks for your suggestion. We have been divided the layer into: Cladding Zone, Transition Zone, Heat Affection Zone. Most of WC particles distributed in the bottom of Cladding Zone, a few of WC particles distributed in the Transition Zone. The microstructure of the TZ and substrate will be research by our group in the future.
Is the determination of NiC3 correct.
Response: Thanks for your question. Our group check the reference and the other data, and consider the determination of Ni3C is correct. The revise of the manuscript have been finished.
Figure 8a. Do you have better quality for this photo? The photo is very good and you can see the friction mechanism, but better quality would give a beautiful SEM photo effect
Response: Thanks for your suggestion. The Fig.8a is the original picture of the line-scanning, we did not have any photo.
You should expand the "Conclusions" section.
Response: Thanks for your suggestion. Our group have been renew the conclusion in the manuscript.

Round 2
Reviewer 1 Report
My comments were well reflected in your revised manuscript. It would be nice if only the following very simple parts could be modified.
1. Line No. 299
Fig. 10d seems to have to be modified to Fig 10b, but it needs to be checked.
2. Line No. 293~300
It would be better if you put the contents of your response as shown below between Line No. 293~300 (or elsewhere).
<A part of previous my comment and your response.>
Line No. 325-327;
At 60N, the coefficient of friction decreases, but why does the wear rate increase? Although the degree of increase of the abrasion rate is decreased compared to the case of 40N.
Response: Thanks for your question. When the load is 60N, the top of the layer was sharply worn. With the progress of the wear test, the WC stick out from the surface of the layer to bear the load, and the load was not enough to destroy WC particles. therefore, when the load was 60N, the wear rate was increase. The debris filled the space between the predominant WC particles and the surface of layer. Therefore, the coefficient of friction was decreased.
Author Response
- Line No. 299
Fig. 10d seems to have to be modified to Fig 10b, but it needs to be checked.
Response: Thanks for your suggestion. Our group check the manuscript and revised the Fig.10dto Fig.7d.
- Line No. 293~300
It would be better if you put the contents of your response as shown below between Line No. 293~300 (or elsewhere).
<A part of previous my comment and your response.>
Line No. 325-327;
At 60N, the coefficient of friction decreases, but why does the wear rate increase? Although the degree of increase of the abrasion rate is decreased compared to the case of 40N.
Response: Thanks for your question. When the load is 60N, the top of the layer was sharply worn. With the progress of the wear test, the WC stick out from the surface of the layer to bear the load, and the load was not enough to destroy WC particles. therefore, when the load was 60N, the wear rate was increase. The debris filled the space between the predominant WC particles and the surface of layer. Therefore, the coefficient of friction was decreased.
Response: Thanks for your suggestion. The contents have been added to the revised manuscript at line No.317.